# The Validation of the 2023 ACR/EULAR Antiphospholipid Syndrome Classification Criteria in a Cohort from Turkey

**DOI:** 10.3390/diagnostics14192205

**Published:** 2024-10-02

**Authors:** Salim Mısırcı, Ali Ekin, Burcu Yağız, Belkıs Nihan Coşkun, Ediz Dalkılıç, Yavuz Pehlivan

**Affiliations:** Department of Internal Medicine, Division of Rheumatology, Faculty of Medicine, Bursa Uludag University, 16059 Bursa, Turkey; aliekin49@hotmail.com (A.E.); burcuyilmaz_84@hotmail.com (B.Y.); belkisnihanseniz@hotmail.com (B.N.C.); edizinci@hotmail.com (E.D.); pehlivany@uludag.edu.tr (Y.P.)

**Keywords:** antiphospholipid syndrome, livedo racemosa, Sapporo criteria, systemic lupus erythematosus

## Abstract

Background/Objectives: Our aim was to validate the performance of the American College of Rheumatology (ACR)/European League Against Rheumatism (EULAR) classification criteria for antiphospholipid syndrome (APS), published in 2023, in an APS cohort. Methods: A total of 193 patients, 83 with APS (secondary APS, *n* = 45; primary APS, *n* = 38) and 110 without APS (systemic lupus erythematosus (SLE), *n* = 100; others, *n* = 10), were included in this study. The performance (sensitivity, specificity and area under the curve (AUC)) of the 2023 ACR/EULAR classification criteria for APS was evaluated and the agreement with the revised Sapporo criteria was compared using the kappa test. Results: In our cohort, the sensitivity and specificity of the 2023 ACR/EULAR classification criteria for APS were 73% and 94%, respectively (AUC: 0.836, 95% CI: 0.772–0.899), while the sensitivity and specificity of the revised Sapporo criteria were 66% and 98%, respectively (95% CI: 0.756–0.888). The performance of the two sets of criteria in our cohort was significantly consistent and significant (*p* < 0.001). When the sensitivity, specificity and ROC curve analysis were performed again by excluding livedo racemosa, the sensitivity of the new criteria in our cohort was 62% and the specificity was 100% (AUC: 0.813, 95% CI: 0.746–0.881). Conclusions: Although the newly published criteria broaden the scope of APS classification by including clinical findings other than thrombosis and obstetric criteria, their sensitivity in our cohort was low. On the other hand, we found that the specificity of the criteria in our cohort reached 100% when livedo findings were excluded.

## 1. Introduction

Antiphospholipid syndrome (APS) is a systemic autoimmune disease characterised by clinical manifestations such as arterial thrombosis (AT), venous thrombosis (VT) and pregnancy morbidity, in which antiphospholipid antibodies (aPL) are positive at least 12 weeks apart [1]. The main aPL antibodies in APS patients are lupus anticoagulant (LAC), anti-cardiolipin (aCL) and anti-β2-glycoprotein-I (aβ2GPI) antibodies. They form a group of antibodies that develop against various negatively charged phospholipids, phospholipid-binding proteins, and phospholipid–protein complex epitopes [2].

APS is divided into secondary APS, where it occurs in the presence of an autoimmune disease such as systemic lupus erythematosus (SLE), and primary APS, where it occurs in the absence of an autoimmune disease [3].

The Sapporo classification criteria for APS were first published in 1999 [4] and revised in 2006 [4]. According to the revised Sapporo criteria, patients with at least one of the clinical signs of thrombosis or pregnancy morbidity can be classified as having APS if persistent aPL positivity is present [4]. Following the publication of the revised Sapporo criteria, the American College of Rheumatology (ACR) and the European League Against Rheumatism (EULAR) published the new ACR/EULAR classification criteria for APS in 2023, resulting from advances in the clinical features of APS apart from thrombosis and pregnancy morbidity [5].

Patients with at least one of the six sites of clinical involvement listed in the 2023 ACR/EULAR APS classification criteria and at least one positive aPL antibody (an LAC test or moderate-to-high titres of aCL or aβ2GPI [IgG or IgM]) within three years of the clinical criteria meet the APS entry criteria. It has been established that patients who meet the APS entry criteria can be classified as having APS if each of the clinical and laboratory criteria has a value of ≥3 by scoring them according to clinical and laboratory involvement [5]. In the revised Sapporo criteria, risk factors for vascular thrombosis and clinical manifestations of APS other than thrombosis and pregnancy are mentioned, but not included in the classification criteria [4]. Among the clinical criteria, thrombosis is assessed according to both the VT and AT risk factors in the 2023 ACR/EULAR APS classification criteria. On the other hand, clinical findings of APS other than thrombosis and obstetric involvement, such as livedo racemosa, livedoid vasculopathy, pulmonary haemorrhage, nephropathy, myocardial disease, adrenal haemorrhage, cardiac valve involvement, and thrombocytopenia, are included in the scoring. Thus, a new classification criterion has been published that takes comprehensive account of the clinical manifestations of APS, that includes individual factors and that is more specific [5].

As there are no published diagnostic criteria for the diagnosis of APS, the diagnosis is made on the basis of clinical and laboratory findings. In our study, we used an APS cohort consisting of patients previously diagnosed with APS by rheumatologists in our clinic based on clinical and laboratory findings. Based on the findings in our patients, we evaluated the performance of the 2023 ACR/EULAR APS criteria and the revised Sapporo classification criteria in clinical practise and compared their agreement.

## 2. Materials and Methods

### 2.1. Study Population

Our study included a total of 193 patients, including 83 patients with APS and 110 without APS, who were treated at Bursa Uludag University Faculty of Medicine Division of Rheumatology, a tertiary care hospital, between August 2013 and August 2023. The no- APS group (100 patients with SLE and 10 patients with other rheumatic diseases) included patients with clinical findings similar to APS, such as thrombocytopenia, livedo, thrombosis, or pregnancy morbidity, but that were not considered to have APS. After approval by the local ethics committee (31 October 2023, 2023-22/8), the patients were retrospectively screened from the hospital’s registration system. The diagnoses of APS, SLE, and other rheumatic diseases were made by an experienced rheumatologist. A total of 83 patients with APS were included in the APS group and 110 patients without APS were included in the no-APS group. APS patients without an additional inflammatory rheumatic disease (IRD) were considered primary APS patients, and APS patients with SLE or another rheumatic disease were considered secondary APS patients. Patients with no clinical or laboratory findings and in whom APS antibodies were not analysed were excluded from this study.

### 2.2. Study Design and Data Collection

The demographic characteristics and laboratory values, including the LAC, aCL, aβ2GPI, antinuclear antibody (ANA), anti-double stranded (ds) DNA, direct Coomb’s positivity and complement component C3 and C4 values, were recorded. ANA and anti-dsDNA were determined using indirect immunofluorescence (IIF), while aPL antibodies (aCL and aβ2GPI) were determined using an enzyme-linked immunosorbent assay (ELISA). A moderate positivity of aPL antibodies required a titre of 40–79 units and a high positivity required a titre of ≥80 units. LAC positivity was determined in a 3-step procedure (screening–mixing study–confirmation) and was assessed as positive or negative. Factors that can lead to a false positive LAC, such as acute thrombosis or the use of anticoagulants, were considered by experienced rheumatologists.

The patients included in this study were assessed for their fulfilment of the 2023 ACR/EULAR APS classification criteria (patients scoring ≥ 3 points on both clinical and laboratory criteria by being evaluated for six clinical and two laboratory criteria) and the revised Sapporo criteria (patients with persistent aPL positivity with at least one of the clinical signs of thrombosis or pregnancy morbidity). The sensitivity and specificity of the 2006 revised Sapporo criteria and the 2023 ACR/EULAR APS classification criteria were analysed. The receiver operating characteristic (ROC) curve was generated using the 2023 ACR/EULAR APS classification criteria. According to the new classification criteria, the ROC analysis was performed by including patients with a total score of ≥6 points, including ≥3 points from clinical criteria and ≥3 points from laboratory criteria. The area under the curve (AUC) was calculated. In addition, sensitivity and specificity were calculated by performing a new assessment without considering the livedo racemosa. Again, an ROC curve was generated and the AUC was calculated for the 2023 ACR/EULAR APS classification criteria. The agreement between the performance of the two criteria in the evaluation of patients in our cohort was assessed.

### 2.3. Statistical Analysis

The compatibility of the continuous variables with a normal distribution was examined using the Kolmogorov–Smirnov and Shapiro–Wilk tests. Continuous variables were expressed as the mean ± standard deviation (SD) or median (minimum–maximum); categorical variables were expressed as *n* (%). As the continuous variables were not normally distributed, the Mann–Whitney U test was used to compare two groups. The chi-squared test was used to compare categorical variables. The kappa (κ) test was performed to assess the agreement between the two classification criteria. The SPSS (Statistical Package for Social Sciences for Windows, version 28.0, IBM Corp, Armonk, NY, USA) package programme was used to calculate the statistical data. *p* < 0.05 was accepted as the statistical significance level.

## 3. Results

A total of 193 patients, including 83 with APS (secondary APS (secondary to SLE) *n* = 45; primary APS, *n* = 38) and 110 without APS (SLE, *n* = 100; others, *n* = 10) were included in this study. Among the patients in the no-APS group (others, *n* = 10) with an IRD diagnosis other than SLE and a history of thrombosis, Behçet’s disease was the most common (*n* = 6). Mixed connective tissue disease (*n* = 1), rheumatoid arthritis (*n* = 2) and Sjogren’s syndrome (*n* = 1) were the other IRDs in patients with a history of thrombosis. The mean age of the patients was 34.40 (±SD: 10.19) in the APS group and 34.35 (±SD: 11.03) in the no-APS group, and was similar in both groups (*p* = 0.804). The female gender was the most common in both groups (*p* = 0.238). In a laboratory evaluation, ANA positivity was more frequent in the no-APS group (93.64%) (*p* < 0.001), while there was no significant difference between the groups for the anti-dsDNA, C3, C4, or direct Coomb’s parameters (*p* > 0.05). The demographic and laboratory parameters of the patients are summarised in Table 1.

The values of the systemic lupus erythematosus disease activity index 2000 (SLEDAI-2K) at the time of the diagnosis of patients with secondary APS (*n* = 45, secondary to SLE) in the APS group and SLE in the no-APS group (*n* = 100) were evaluated and compared. No statistically significant differences were found between the groups (*p* = 0.087; secondary APS group: median = 10 (6, 22), no-APS group: median = 8 (6, 22)).

### 3.1. Clinical and Laboratory Parameters of APS and No-APS Patients according to the 2023 ACR/EULAR Antiphospholipid Syndrome Classification Criteria

#### 3.1.1. Clinical Parameters of APS and no-APS Patients according to the 2023 ACR/EULAR Antiphospholipid Syndrome Classification Criteria

According to the 2023 ACR/EULAR APS classification criteria, there were nine (10.84%) patients in the APS group who did not meet the entry criteria (one clinical criterion and one positive aPL test). Four of these patients were admitted as APS patients because, at the time of their diagnosis, they had recurrent VT that could not be explained by other causes, but they did not meet the entry criteria because their aPL test was negative. The other five patients were accepted as APS patients because of positive aPL tests and recurrent pregnancy losses at the time of their diagnosis, but they did not meet the entry criteria because they experienced < 3 consecutive losses before 10 weeks of gestation. When the patients were evaluated using the 2023 ACR/EULAR APS classification criteria, thrombocytopenia was the most common clinical criterion among the six main clinical criteria in both groups, and no significant differences were found between the groups (APS group, *n* = 42, 50.60%; no-APS group, *n* = 49, 44.55%; *p* = 0.404). The second most common clinical criterion in the APS group was the VT criterion of macrovascular involvement (*n* = 35, 42.17%), while in the no-APS group, it was microvascular involvement (livedo racemosa) (*n* = 22, 26.51%). The least frequent clinical criterion among the six clinical criteria in the APS group was the cardiac criterion, which was found in only three patients (3.61%). All three of our patients were assessed on the basis of the results of their previous echocardiography. Adrenal haemorrhage (imaging or pathology), myocardial disease (imaging or pathology) and pulmonary haemorrhage (bronchoalveolar lavage or pathology) among the established involvements within the microvascular clinical criteria in APS patients were not seen in any patients, while livedoid vasculopathy involvement (pathology) was found in only one patient (1.20%). If we look again at the suspected involvement in the microvascular criteria in the APS group, livedo racemosa (exam) was found most frequently (*n* = 21, 25.30%), while acute/chronic aPL-nephropathy (exam or lab) was found in only one patient (1.20%). The patient assessed as positive for acute/chronic aPL-nephropathy (exam or lab) had primary APS; a renal ultrasound was performed at the time of diagnosis and postrenal causes were excluded. Livedoid vasculopathy lesions (exam) and pulmonary haemorrhage (symptoms and imaging) were not detected in any patient. Among the 70 female patients in the APS group (84.34%) who met the obstetric criteria (*n* = 23, 32.86%), the most common sub-criteria were >3 consecutive pre-foetal deaths (<10 w) and/or one early foetal death (10 w 0 d–15 w 6 d). The obstetric sub-criteria pre-eclampsia (PEC) with severe features (<34 w 0 d) and placental insufficiency (PI) with severe features (<34 w 0 d) with/without foetal death were only found in one patient (4.35%).

#### 3.1.2. Laboratory Parameters of APS and no-APS Patients according to the 2023 ACR/EULAR Antiphospholipid Syndrome Classification Criteria

When patients were evaluated using laboratory criteria, the most common aPL antibody detected in both groups was LAC, which was positive in 65 (78.31%) patients in the APS group and 24 (21.82%) patients in the no-APS group (*p* < 0.001). When the persistent aCL and/or aβ2GPI positivity of the patients in the no-APS group was evaluated in detail, aCL-IgM was positive in two patients and the titres were 60 and 120 units, respectively. aCL-IgG was positive in six patients and the titres were 45, 65 and 70 units in three patients, respectively, while the titres were >120 in all three of the other patients. Both aCL-IgG and aβ2GPI-IgG were positive in three patients; the aCL-IgG titres were >120 in all of these patients; and the aβ2GPI-IgG titres were 80, 100, and 120 units, respectively. The clinical and laboratory criteria of patients in the APS and no-APS groups according to the 2023 ACR/EULAR APS classification criteria are summarised in Table 2.

### 3.2. The Sensitivity and Specificity of the 2023 ACR/EULAR Antiphospholipid Syndrome (APS) Classification Criteria Compared to the Revised Sapporo APS Classification Criteria in Our Cohort

The sensitivity and specificity of the 2023 ACR/EULAR APS classification criteria for APS in our cohort were 73% and 94%, respectively, with an AUC of 0.836 (95% CI: 0.772–0.899). The sensitivity and specificity of the revised Sapporo criteria were 66% and 98%, respectively (95% CI: 0.756–0.888). When the agreement in the performance of the two classification criteria was evaluated, the κ value was 0.752 (*p* < 0.001) and they were found to be statistically significant and significantly compatible. The assessments of sensitivity and specificity as well as the analysis of the ROC curve can be found in Figure 1 and Table 3. As shown in Table 3, there were 22 patients in our cohort who had previously been diagnosed and followed up with for APS, but who did not meet the 2023 ACR/EULAR APS classification criteria. When we analysed these patients separately, four patients had recurrent VT without a high-risk VTE profile, but the criteria were not met because the aPL tests were negative. Five patients met the clinical criteria due to thrombosis (*n* = 2, VTE without high-risk profile; *n* = 3, AT without high-risk CVD profile), but the laboratory criteria were not met due to a single positive LAC test. A further five patients had positive aPL tests (*n* = 5, aCL-IgG positive at high titre) and, although the laboratory criteria were met, the clinical criteria were not met due to <3 consecutive losses before 10 weeks of gestation. In the other eight patients, the aPL tests were positive (*n* = 4, LAC persistent positive; *n* = 2, aCL-IgG positive at high titre; *n* = 2, aβ2GPI-IgG positive at high titre) and the laboratory criteria were met, but the clinical criteria were not met due to a score of 1 according to the newly published criteria (at least 3 points are required), despite ≥3 consecutive losses before 10 weeks of gestation.

### 3.3. The Sensitivity and Specificity of the 2023 ACR/EULAR Antiphospholipid Syndrome (APS) Classification Criteria (Without Livedo Racemosa) Compared to the Revised SAPPORO APS Classification Criteria in Our Cohort

The group without APS included seven patients who had not previously been recognised as having APS by the rheumatologists at our clinic, but who met the ACR/EULAR APS classification criteria of 2023. In previous evaluations of these patients by experienced rheumatologists, the aPL antibodies were positive and they met the laboratory criteria, but they were not recognised as having APS because they had no history of thrombosis or significant obstetric findings consistent with APS. Seven patients met the clinical criteria due to thrombocytopenia and livedo racemosa findings. Of the six clinical criteria of the 2023 ACR/EULAR APS classification criteria, livedo racemosa with microvascular involvement was excluded from the scoring and sensitivity, specificity and ROC curve analyses were re-performed (Figure 2 and Table 4). In the new evaluation, the sensitivity and specificity of the 2023 ACR/EULAR APS classification criteria for APS in our cohort were 62% and 100% (AUC: 0.813, 95% CI: 0.746–0.881), while the sensitivity and specificity of the revised Sapporo criteria were 66% and 98% (95% CI: 0.756–0.888), respectively. When evaluating the agreement in the performance of the two classification criteria, the kappa value was 0.911 (*p* < 0.001) and they were found to be statistically significant and highly concordant.

## 4. Discussion

In our study, we examined the performance of the newly published 2023 ACR/EULAR APS classification criteria in an APS cohort and a no-APS cohort in a tertiary care hospital. We also found that the specificity of the new criteria increased from 94% to 100% when we excluded the livedo racemosa criterion from microvascular involvement in the clinical criteria. However, when the agreement between the new classification criteria and the revised Sapporo classification criteria was assessed in our cohort, the performance of both criteria was statistically significant and significantly concordant.

In addition to 100 SLE patients with clinical features similar to APS (thrombocytopenia, livedo, thrombosis, or pregnancy morbidity), 10 IRD patients with a history of thrombosis were included in the group without APS and the scope of the study was expanded. On the other hand, although the majority of the patients in the no-APS group had SLE, it is difficult and important to differentiate secondary APS patients from SLE patients, mainly because of the similar clinical findings.

SLE and APS occur more frequently in young and middle-aged people [6,7]. In a study of 1000 patients with APS, the average age was 42 years and 82% of the patients were female. Primary APS was observed in about half of the patients [7]. In our study, the average age of the patients in the APS group was 34.4 years, the proportion of women was 70%, and the rate of primary APS was 45.78%, which is consistent with the literature.

In the laboratory evaluation, ANA positivity was significantly higher in the no-APS group. Although ANA positivity is not a specific test for the diagnosis of SLE, it is very sensitive and its positivity is required in the 2019 ACR/EULAR classification criteria for patients to be classified as having SLE [8]. Although ANA positivity was higher in the no- APS group, the main factor resulting in the detection of ANA with a rate of 75% in the APS group was the SLE patients in the secondary APS group. Again, the main reason for the lack of a difference between the groups in the anti-dsDNA, C3, C4 and direct Coomb’s parameters seemed to be that patients with secondary APS (secondary to SLE) predominated in the APS group.

In the newly published criteria, patients must first fulfil the entry criteria (≥1 documented clinical criterion + ≥1 positive aPL test) to be classified as having APS [5]. Of the nine (10.84%) patients in the APS group who did not meet the entry criteria, the aPL test was negative in four and the clinical criterion (<3 consecutive losses (<10 w)) was not met in five. Patients with clinical features for APS, but who are negative for LAC, aCL and aβ2GPI (antibodies that are routinely checked and represent laboratory criteria) can be considered to have seronegative APS. One of the factors for these results is that APS-associated antibodies other than LAC, aCL and aβ2GPI were not tested [9]. Anti-prothrombin (aPT) and the anti-phosphatidylserine/prothrombin complex (aPS/PT) are non-criteria antibodies, and aPS/PT may be especially associated with thrombosis [10]. The global antiphospholipid syndrome score (GAPSS), constructed in combination with cardiovascular risk factors and aPL antibodies in the criteria, was found to be more effective at predicting the clinical findings associated with aPS when aPS/PT positivity was included in the assessment [11]. In a study on non-criteria aPLs, aPS/PT, anti-annexin V, anti-phosphatidylethanolamine (aPE), aPS and aPhL (a mixture of negatively charged phospholipids) antibodies were evaluated. It was emphasised that the non-criteria aPLs are important for increasing the diagnostic accuracy and predicting the risk of thrombosis in APS [12]. In our study, we think that there may have been untested aPL antibodies in our four patients who were assumed to have APS, but whose aPL antibodies were negative.

In a cohort evaluating non-criteria obstetric findings, two spontaneous abortions (<10 w) in 16 patients were the most common findings among non-criteria obstetric findings (infertility, two or more in vitro fertilisation failures, intrauterine growth retardation and prematurity) [13]. In our study, aPL antibodies were positive in five patients in the APS group, although they did not fully meet the obstetric criteria for APS, and no other reason could be found to explain the abortions.

VT is more common in APS patients than AT. However, AT is more life-threatening. In studies, the rates of VT vary between 35.7% and 87.4% [14]. The results of our study are consistent with the literature.

In the newly published criteria, microvascular involvement, thrombocytopenia as a haematological criterion and cardiac valve involvement were added to the criteria, with the exception of thrombosis and obstetric criteria [5]. In a large cohort of 1000 patients with APS, thrombocytopenia and livedo reticularis were found in 21.9% and 20.4% of patients, respectively. Cardiac valve involvement was found in 11.6% of patients at follow-up [15]. In another study examining the clinical features of aPL-positive patients participating in the APS Alliance for Clinical Trials and International Networking (APS ACTION), cardiac valve involvement was found in 10%, thrombocytopenia in 19% and livedo reticularis/racemosa in 14%. In this study, livedo reticularis and livedo racemosa were not differentiated [16]. In our study, a lower involvement of the cardiac valves (3.61%) was found compared to the literature. On the other hand, not all of our patients had echocardiographic findings, which may have been a factor in obtaining the present results. Thrombocytopenia was found in 50.6% of the patients in the APS group, and our results differ from those in the literature. The presence of thrombocytopenia in patients is defined in the literature as platelet counts <100 × 10^9^/L [15,16]. However, the definition of thrombocytopenia (lowest 20–130 × 10^9^/L) for the haematological criteria is different in the newly published criteria [5]. The higher rate of thrombocytopenia in our APS group compared to the literature could be related to this condition. Livedo reticularis is a transient or irreversible, reddish-blue-purple reticular skin finding that is usually benign and usually occurs in middle-aged women. On the other hand, livedo racemosa is an irreversible pathological skin finding with tears in the reticular structure and is often associated with APS [17]. Livedo reticularis is not included in the scoring in the newly published criteria due to its low specificity for APS. The livedo racemosa finding, which is a pathologic finding, is included in the scoring [5].

In our study cohort, we found that the newly published criteria for APS had a higher sensitivity (73% and 66%, respectively) but a lower specificity (94% and 98%, respectively) compared with the revised Sapporo criteria. In contrast, when a new analysis was performed without the livedo racemosa finding, the specificity of the new criteria was better than that of the revised Sapporo criteria (100% and 98%, respectively). This specificity result (100%) was similar to the specificity in the 2023 ACR/EULAR APS classification criteria validation cohorts (99% in both). Our decision to re-evaluate without livedo racemosa was due to the high rate of livedo racemosa in the clinical findings of the patients in our cohort. This could be due to the fact that the livedo reticularis findings, which are benign findings that can occur in IRDs such as SLE in addition to APS, were not fully differentiated from livedo racemosa findings and were recorded as livedo racemosa. The sensitivity of the new criteria was lower in our cohort (73% and 62% (without livedo racemosa)) than in the 2023 ACR/EULAR APS classification criteria validation cohorts (83% and 84%, respectively), mainly due to seronegative APS patients (*n* = 4), non-persistent positivity of the LAC test (*n* = 5) and patients with obstetric involvement who did not meet the newly defined clinical criteria score. When we analysed the five patients with non-persistent LAC tests, we found that anticoagulant treatment was initiated after the clinical findings and LAC positivity, and the control LAC test was not accepted as persistently positive because it was positive under anticoagulants. In the scoring system of the new criteria, given the one point assigned for ≥3 consecutive losses before 10 weeks’ gestation, the fact that this score alone is not sufficient (at least three points are required) was one of the main factors that decreased the sensitivity in our cohort. In addition, the fulfilment of the new criteria in patients (*n* = 7) in the no-APS group, in whom thrombocytopenia (two points) and livedo racemosa (two points) were detected, led to a decrease in specificity.

On the other hand, the performance of the newly published 2023 ACR/EULAR APS criteria and the revised Sapporo criteria were significantly compatible in our cohort. The main factors for these results seem to be the more difficult fulfilment of the new criteria due to the above-mentioned assessment of obstetric involvement and the inclusion of findings such as thrombocytopenia and livedo racemosa, which were not included in the Sapporo criteria, in the new criteria. The fact that both criteria worked well and were compatible in our cohort could raise the question of what additional contribution the new criteria make compared to the revised Sapporo criteria. However, the inclusion of other clinical findings that may occur in the course of APS besides thrombosis and obstetric involvement in the new classification criteria may be important in order for rheumatologists to focus on other clinical findings. Although it is recognised that classification criteria should only be used for classification, they are also frequently used as diagnostic criteria. According to our study, the sensitivity of the new criteria in our cohort (73% and 62% (without livedo racemosa)) was low, and therefore, the limitations of the criteria should be considered when making a diagnosis. On the other hand, the specificity of the new criteria reached 100% when the livedo racemosa finding was not considered.

### Limitations

The main limitation of our study is that we retrospectively evaluated patients who were recognised as having APS by a specialist rheumatologist in the rheumatology clinic of a single tertiary hospital. This may lead to criticism. The other limitations of the study include the small number of patients and the lack of a separate sub-analysis for patients with primary APS and secondary APS. Another important limitation is that the livedo findings could not be assessed by re-examining the livedo reticularis and livedo racemosa due to the retrospective design of the study.

## 5. Conclusions

We validated the newly published ACR/EULAR APS classification criteria of 2023 in our cohort. We also found that the specificity of the criteria reached 100% when livedo findings were excluded. When the performance of the new classification criteria and the revised Sapporo classification criteria were evaluated in our cohort, both sets of criteria showed significant and consistent performance. However, the newly published criteria have made it possible to classify patients as having APS by addressing, in detail, findings other than thrombosis and the obstetric criteria of APS, thus allowing rheumatologists to focus on these findings. However, clinicians should exercise particular caution when assessing livedo findings and obstetric involvement. On the other hand, seronegative APS patients, and, thus, aPL antibodies other than the APS antibodies that are routinely assessed and used in the criteria, appear to remain the subject of research.

## Figures and Tables

**Figure 1 diagnostics-14-02205-f001:**
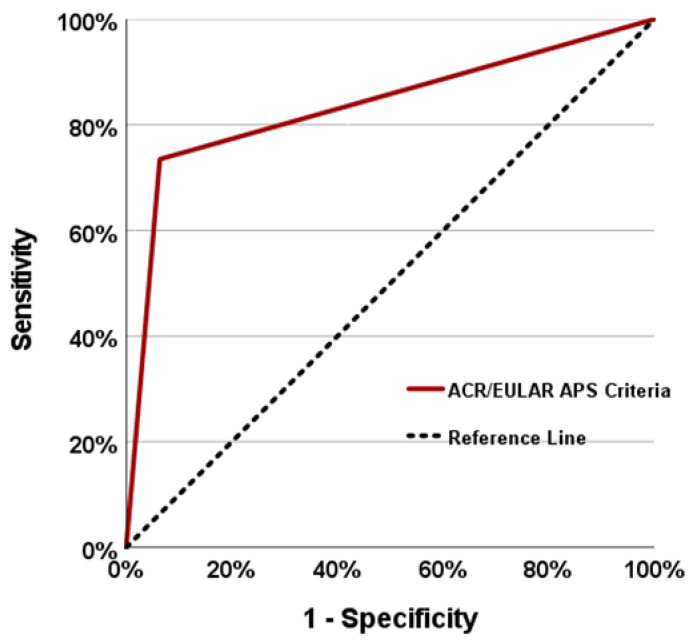
Receiver operating characteristic (ROC) curve of the ACR/EULAR antiphospholipid syndrome (APS) classification criteria.

**Figure 2 diagnostics-14-02205-f002:**
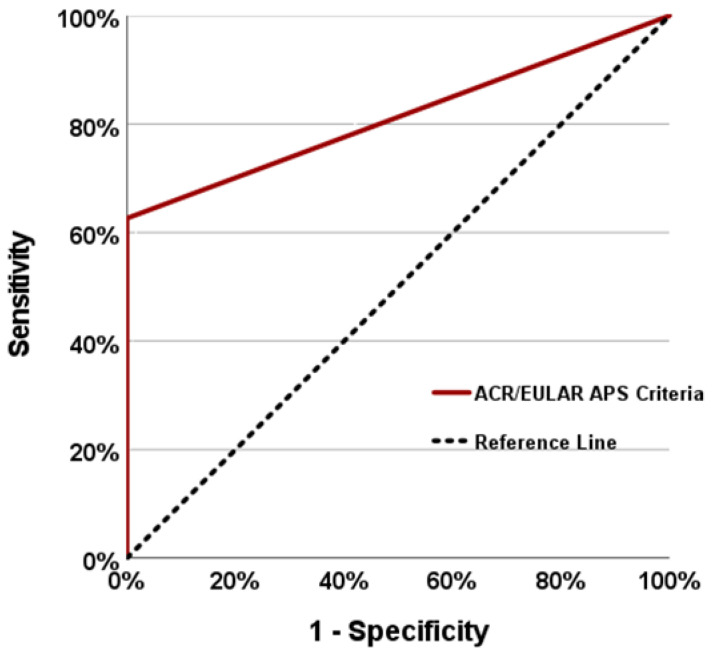
Receiver operating characteristic (ROC) curve of the ACR/EULAR antiphospholipid syndrome (APS) classification criteria (without livedo racemosa).

**Table 1 diagnostics-14-02205-t001:** Demographic and laboratory parameters of APS and no-APS patients.

	APS *n* (%) 83	No-APS *n* (%) 110	*p*
Age (year)	34.40 (±SD: 10.19)	34.35 (±SD: 11.03)	0.804 ^m^
Sex			0.238 ^χ2^
Female	70 (84.34)	99 (90.00)	
Male	13 (15.66)	11 (10.00)	
ANA positivity	63 (75.90)	103 (93.64)	**<0.001 ^χ2^**
Anti-dsDNA	30 (36.14)	33 (30.00)	0.367 ^χ2^
C3 low	31 (37.35)	42 (38.18)	0.906 ^χ2^
C4 low	20 (24.10)	34 (30.91)	0.297 ^χ2^
Direct Coomb’s	26 (31.33)	34 (30.91)	0.951 ^χ2^

ANA, antinuclear antibody; Anti-dsDNA, anti-double stranded DNA antibodies; C3, complement component 3; C4, complement component 4; ^m^, Mann–Whitney U test; χ2, chi-squared test; *p* < 0.05: statistical significance level.

**Table 2 diagnostics-14-02205-t002:** Clinical and laboratory parameters of APS and no-APS patients according to the 2023 ACR/EULAR antiphospholipid syndrome classification criteria.

	APS *n* (%) 83	No-APS *n* (%) 110	*p*
**Entry criteria**			**<0.001 ^χ2^**
Yes	74 (89.16)	22 (20.00)	
No	9 (10.84)	88 (80.00)	
**Clinical Domains and Criteria**			
**Macrovascular (VTE)**	35 (42.17)	18 (16.36)	**<0.001 ^χ2^**
VTE with a high-risk VTE profile	3 (3.61)	2 (1.82)	0.437 ^χ2^
VTE without a high-risk VTE profile	32 (38.55)	16 (14.55)	**<0.001 ^χ2^**
**Macrovascular (AT)**	28 (33.73)	6 (5.45)	**<0.001 ^χ2^**
AT with a high-risk CVD profile	6 (7.23)	2 (1.82)	0.062 ^χ2^
AT without a high-risk CVD profile	22 (26.51)	4 (3.64)	**<0.001 ^χ2^**
**Microvascular**	23 (27.71)	20 (18.18)	0.115 ^χ2^
**Suspected**	22 (26.51)	20 (18.18)	0.165 ^χ2^
Livedo racemosa (exam)	21 (25.30)	19 (17.27)	0.173 ^χ2^
Livedoid vasculopathy lesions (exam)	0 (0.00)	1 (0.91)	>0.05 ^χ2^
Acute/chronic aPL-nephropathy (exam or lab)	1 (1.20)	0 (0.00)	0.430 ^χ2^
**Established**: Livedoid vasculopathy (pathology)	1 (1.20)	0 (0.00)	0.430 ^χ2^
**Obstetric, no/total**	23/70 (32.86)	3/99 (3.03)	**<0.001 ^χ2^**
>3 Consecutive pre-foetal (<10 w) and/or early foetal (10 w 0 d–15 w 6 d) deaths	17/23 (73.91)	2/99 (2.02)	**<0.001 ^χ2^**
Foetal death (16 w 0 d–33 w 6 d) in the absence of PEC with severe features or PI with severe features	3/23 (13.04)	1/99 (1.01)	0.316 ^χ2^
PEC with severe features (<34 w 0 d) or PI with severe features (<34 w 0 d) with/without foetal death	3/23 (13.04)	0/99 (0.00)	0.078 ^χ2^
PEC with severe features (<34 w 0 d) and PI with severe features (<34 w 0 d) with/without foetal death	1/23 (4.35)	0/99 (0.00)	0.430 ^χ2^
**Cardiac**	3 (3.61)	0 (0.00)	0.078 ^χ2^
Valve Thickening	1 (1.20)	0 (0.00)	0.430 ^χ2^
Vegetation	2 (2.41)	0 (0.00)	0.184 ^χ2^
**Haematology**: Thrombocytopenia (lowest 20–130 × 10^9^/L)	42 (50.60)	49 (44.55)	0.404 ^χ2^
**Laboratory Domains and Criteria**			
**Lupus anticoagulant-positive**	65 (78.31)	24 (21.82)	**<0.001 ^χ2^**
Positive (single-once)	25 (30.12)	20 (18.18)	0.052 ^χ2^
Positive (persistent)	40 (48.19)	4 (3.64)	**<0.001 ^χ2^**
**aCL and/or aβ2GPI (** **persistent) *****	53 (63.86)	11 (10.00)	**<0.001 ^χ2^**
Moderate or high positive (IgM)(aCL and/or aβ2GPI)	10 (12.05)	2 (1.82)	**0.004 ^χ2^**
Moderate positive (IgG) (aCL and/or aβ2GPI)	15 (18.07)	3 (2.73)	**<0.001 ^χ2^**
High positive (IgG) (aCL or aβ2GPI)	14 (16.87)	3 (2.73)	**<0.001 ^χ2^**
High positive (IgG) (aCL and aβ2GPI)	14 (16.87)	3 (2.73)	**<0.001 ^χ2^**

aβ2GPI, anti-β2-glycoprotein-I antibody; aCL, anti-cardiolipin antibody; aPL, anti-phospholipid; APS, antiphospholipid syndrome; AT, arterial thrombosis; CVD, cardiovascular disease; PEC, pre-eclampsia; PI, placental insufficiency; VTE, venous thromboembolism; χ2, chi-square test; *, moderate-level (40–79 units) and high-level (≥80 units) aCL/anti-β2GPI are based on enzyme-linked immunosorbent assays; *p* < 0.05: statistical significance level.

**Table 3 diagnostics-14-02205-t003:** The sensitivity and specificity of the 2023 ACR/EULAR antiphospholipid syndrome (APS) classification criteria compared to the revised Sapporo APS classification criteria in our cohort.

	APS	No-APS	APS	No-APS
	(*n* = 83)	(*n* = 110)	(*n* = 83)	(*n* = 110)
	2023 ACR/EULAR APS Criteria		Revised Sapporo APS Criteria	
Criteria				
Yes	61	7	55	2
No	22	103	28	108
Sensitivity	73%		66%	
Specificity	94%		98%	
AUC (95% CI)	0.836 (0.772–0.899)		(0.756–0.888)	

APS, antiphospholipid syndrome; 95% CI, 95% Confidence Interval.

**Table 4 diagnostics-14-02205-t004:** The sensitivity and specificity of the 2023 ACR/EULAR antiphospholipid syndrome (APS) classification criteria (without livedo racemosa) compared to the revised Sapporo APS classification criteria in our cohort.

	APS	No-APS	APS	No-APS
	(*n* = 83)	(*n* = 110)	(*n* = 83)	(*n* = 110)
	2023 ACR/EULAR APS Criteria		Revised Sapporo APS Criteria	
Criteria				
Yes	52	0	55	2
No	31	110	28	108
Sensitivity	62%		66%	
Specificity	100%		98%	
AUC (95% CI)	0.813 (0.746–0.881)		(0.756–0.888)	

APS, antiphospholipid syndrome; 95% CI, 95% Confidence Interval.

## Data Availability

The data presented in this study are available on request from the corresponding author.

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
