# Peer review of "The Validation of the 2023 ACR/EULAR Antiphospholipid Syndrome Classification Criteria in a Cohort from Turkey"

_diagnostics, 2024, doi:10.3390/diagnostics14192205_

Round 1

Reviewer 1 Report

Comments and Suggestions for Authors

This is a validation of the new ACR /EULAR 2023 APS criteria in a single turkish cohort of patients with a total of 193 patients, 83 with APS (secondary 11 APS n=45, primary APS n=38) and 110 no APS (systemic lupus erythematosus (SLE) n=100, others  n=10) were included in the validation study.

Major Comments:

Your validation is actually between Primary APS, Secondary APS and Lupus patients, since the minority of your non APS patients had other diseases than Lupus. Considering that the validation of the ACR EULAR 2023 APS Criteria has included patients with secondary APS, I consider ok to do this comparison but, the authors should alert the readers regarding this since there is a chance that SLE and APS manifestations might be hard to be separated in the secondary APS group.

 It very unusual that patients with primary APS are anti-dsDNA +. More than 30% of your cohort were antidsDNA +. Please comment.

Can you please provide details regarding the antidsDNA methodology? What about the aPL antibodies? Including titers range for anti-aCL and anti-B2GPI?

Please add informations regarding the SLEDAIs of your lupus patients. It is very important especially considering the secondary APS group and SLE group.

“The least frequent clinical criterion among the six clinical criteria in the APS group was the cardiac criteria, which was found in only three patients (3.61%). Did you actively checked your patients with echos? Or did you report what was historically found in the files?

“while acute/chronic aPL-nephropathy (exam or lab) was found in only one patient (1.20%)”. The same question above for nephropathy, how did you include these patients? Do your patients have done kidney ultrasound? Nephropathy in APS is frequently underreported since it is more difficult to biopsy patients who are under anticoagulation.

Can you provide the aPL titers, isotypes and LA positivity, for the patients without diagnosis of APS?

“In the laboratory evaluation, ANA positivity was significantly higher in the no APS group.” For making this comparison the authors would need to eliminate the secondary APS group from the APS group, since in this group there are also 50% of patients with SLE.

“Of the nine(10.84%) patients in the APS group who did not meet the entry criteria, the aPL test was negative in four and the clinical criterion (< 3 consecutive losses (< 10w)) was not met in five.” Can you please make it clear which is the clinical criteria for these patients?

“than in the 2023 ACR/EULAR APS classification criteria validation cohorts (83% and 84%, respectively), mainly due to seronegative APS patients (n = 4), non-persistent positivity of the LAC test (n = 5), and patients with obstetric involvement who did not meet the newly defined clinical criteria score. Can you please specifiy separetely the clinical and laboratory criteria that these patients have to be classified as APS in your cohort?

“This could be due to the fact that the livedo reticularis findings, which is a benign finding that can occur in IRDs such as SLE in addition to APS, were not fully differentiated from livedo racemose findings and were recorded as livedo racemose.”. This should be considered as a major limitation of your study especially considering that you want to propose that the criteria perform better if racemosa is excluded. For well trained researches / doctors it is easy to differentiate livedo reticularis from racemosa.

MInor Comment:

“the revised Sapparo criteria”- typo Sapporo, adjust it in many spots in your text.

“livedo racemose”- typo racemosa, adjust it in many spots in your text.

because < 3 consecutive losses (< 10w). Please complete the whole sentence.

“In our study, a lower involvement of the cardiac valves (3.61%) was found compared to the literature”. Can you please make it clear?

Reviewer 2 Report

Comments and Suggestions for Authors

This is a nice study entitled The Validation of the 2023 ACR/EULAR Antiphospholipid Syndrome Classification Criteria in a Cohort from Turkey.  This important issue has been properly addressed although some limitation should be highlighted and addressed:

a.     Although classification criteria should be used for classification only, they are commonly used as diagnostic criteria. According to this study the sensitivity was 62%-73% hence the limitation for diagnosis should be highlighted.  

b.     The assessment of these criteria with deletion of livedo racemose seems reasonable but should be explained. Notably this important manifestation is rarely documented and there is lack of knowledge on its definition.

c.     The comparison of the classification criteria is not clear -  how ? to what the 2006 and 2023 criteria were compared ? if understood  correctly this is to historical clinical diagnosis ? the difference between clinical diagnosis as the gold standard vs. classification criteria - should be clarified

d.     Some limitation should be emphasized:

a.     The cohort size is small

b.     There is no sub-analysis to primary vs. secondary APS which is of importance

Minor comment

Please note that the name of the institute line 68 is missing (method section)

Reviewer 3 Report

Comments and Suggestions for Authors

This study comprises an interesting further validation of the 2023 ACR/EULAR Antiphospholipid Syndrome criteria in a cohort from Turkey. The study is designed appropriately to address the issue at stake. However, the most striking finding here is that the new criteria did not perform any better than the revised Sappora criteria. For example, the AUCs of the ROC curves appear to not differ at all among the 2 criteria sets. A statistical comparison should be included. The differences in sensitivity and specificity are marginal and hence most likely clinically irrelevant (e.g. 98% versus 100% for specificity). I believe that this issue requires a paragraph that deals with this in the discussion. The differences in ROC, sensitivities and specificities should also be given in the abstract.

It would be of interest to know what the diagnoses were in the 10 ‘additional IRD patients’.

English language editing is needed. I do not understand what ‘could not be controlled’ in line 298 means.  

Comments on the Quality of English Language

Moderate editing needed. 

Round 2

Reviewer 1 Report

Comments and Suggestions for Authors

All questions were properly addressed by the authors.

Author Response

Comment: All questions were properly addressed by the authors.

Response:

Dear Reviewer 1,

We are pleased that the revisions we have made have been accepted by you. We are grateful for your suggestions for improving our article.

Sincerely

Dr. Salim Mısırcı

Reviewer 3 Report

Comments and Suggestions for Authors

I still cannot find a statistical analysis that compares the AUCs of the ROC curves. I believe that this is required and highly relevant in the present context and, further, it should be addressed adequately in the Discussion. The lines that are referred to in the replies to my comments are mostly incorrect.

Author Response

First of all, we would like to thank you for your valuable advice on correcting the content of our article. We have made the corrections in line with your suggestions. Since the results of sensitivity and specificity of the two sets of criteria in our cohort were close to each other and in accordance with your wishes, we obtained the support of the Department of Biostatistics at our university for the statistical comparison. It was determined that it would be appropriate to use the kappa (κ) test to assess the performance of the two sets of criteria in our cohort and that it would be sufficient to report the ROC curve for ACR/EULAR. In line with these suggestions, the κ-test was used in the statistics section. The adjustments were made in Figure 1, Figure 2, Table 3 and Table 4. As you highlighted, both sets of criteria performed well in our cohort as a result of the κ-test. The concordant performance of the two criteria in our cohort is emphasised by making adjustments in both the discussion and the conclusion. In addition, the old suggested revisions have been rewritten in the new revision to correct the incorrect line numbers in the previous revision.

We look forward to hearing from you if you have any further correction requests.

Sincerely

Dr. Salim Mısırcı

Comment: I still cannot find a statistical analysis that compares the AUCs of the ROC curves. I believe that this is required and highly relevant in the present context and, further, it should be addressed adequately in the Discussion. The lines that are referred to in the replies to my comments are mostly incorrect.

Response : We have changed various parts of the article according to your suggestions.

Abstract

(lines:13-14-15) (The performance (sensitivity, specificity and area under the curve (AUC)) of the 2023 ACR/EULAR classification criteria for APS was evaluated and the agreement with the revised Sapporo criteria was compared using the kappa test.)

(lines:18-19) (The performance of the two sets of criteria in our cohort was significantly consistent and significant (p<0.001).)

Statistical analysis

(lines:117-118) (The kappa (κ) test was performed to assess the agreement between the two classification criteria.)

Results

3.2. The sensitivity and specificity of the 2023 ACR/EULAR antiphospholipid syndrome (APS) classification criteria compared to the revised Sapporo APS classification criteria in our cohort

(lines: 205-206-207) (When the agreement in the performance of the two classification criteria was evaluated, the κ value was 0.752 (p < 0.001) and they were found to be statistically significant and significantly compatible.)

Results

3.3. The sensitivity and specificity of the 2023 ACR/EULAR antiphospholipid syndrome (APS) classification criteria (without livedo racemosa) compared to the revised Sapporo APS classification criteria in our cohort

(lines:244-245-246-247) (When evaluating the agreement in the performance of the two classification criteria, the kappa value was 0.911 (p < 0.001) and they were found to be statistically significant and highly concordant.)

Discussion

First paragraph (lines:260-261-262) (However, when the agreement between the new classification criteria and the revised Sapporo classification criteria was assessed in our cohort, the performance of both criteria was statistically significant and significantly concordant.)

(lines:345-346-347-348-349-350-351-352-353-354-355-356-357-358) (The sensitivity of the new criteria was lower in our cohort (73% and 62% (without livedo racemosa)) than in the 2023 ACR/EULAR APS classification criteria validation cohorts (83% and 84%, respectively), mainly due to seronegative APS patients (n = 4), non-persistent positivity of the LAC test (n = 5), and patients with obstetric involvement who did not meet the newly defined clinical criteria score. When we analysed the five patients with non-persistent LAC tests, we found that anticoagulant treatment was initiated after the clinical findings and LAC positivity, and the control LAC test was not accepted as persistently positive because it was positive under anticoagulants. In the scoring system of the new criteria, one point for ≥ 3 consecutive losses before 10 weeks' gestation and the fact that this score alone is not sufficient (at least three points are required) was one of the main factors that decreased the sensitivity in our cohort. In addition, the fulfilment of the new criteria in patients (n=7) in the no-APS group, in whom thrombocytopenia (two points) and livedo racemosa (two points) were detected, led to a decrease in specificity.)

The last paragraph of the "Discussion" section has been restructured. (lines:359-360-361-362-363-364-365-366-367-368-369-370-371-372-373-374)

(On the other hand, the performance of the newly published 2023 ACR/EULAR APS criteria and the revised Sapporo criteria were significantly compatible in our cohort. The main factors for these results seem to be the more difficult fulfilment of the new criteria due to the above mentioned assessment of obstetric involvement and the inclusion of findings such as thrombocytopenia and livedo racemosa, which were not included in the Sapporo criteria, in the new criteria. The fact that both criteria worked well and were compatible in our cohort could raise the question of what additional contribution the new criteria make compared to the revised Sapporo criteria. However, the inclusion of other clinical findings that may occur in the course of APS besides thrombosis and obstetric involvement in the new classification criteria may be important in order for rheumatologists to focus on other clinical findings. Although it is recognised that classification criteria should only be used for classification, they are also frequently used as diagnostic criteria. According to our study, the sensitivity of the new criteria in our cohort (73% and 62% (without livedo racemosa)) was low, and therefore, the limitations of the criteria should be considered when making a diagnosis. On the other hand, the specificity of the new criteria reached 100% when the livedo racemosa finding was not considered.)

Conclusion

The "Conclusion" has been restructured according to your suggestions in the "Discussion" section.

(lines:386-387-388-389-390-391-392) (When the performance of the new classification criteria and the revised Sapporo classification criteria were evaluated in our cohort, both sets of criteria showed significant and consistent performance. However, the newly published criteria have made it possible to classify patients as having APS by addressing, in detail, findings other than thrombosis and the obstetric criteria of APS, thus allowing rheumatologists to focus on these findings. However, clinicians should exercise particular caution when assessing livedo findings and obstetric involvement.)

The line numbers of the corrections relating to your earlier suggestions have been rearranged.

Comment 1: This study comprises an interesting further validation of the 2023 ACR/EULAR Antiphospholipid Syndrome criteria in a cohort from Turkey. The study is designed appropriately to address the issue at stake. However, the most striking finding here is that the new criteria did not perform any better than the revised Sappora criteria. For example, the AUCs of the ROC curves appear to not differ at all among the 2 criteria sets. A statistical comparison should be included. The differences in sensitivity and specificity are marginal and hence most likely clinically irrelevant (e.g. 98% versus 100% for specificity). I believe that this issue requires a paragraph that deals with this in the discussion.

Response 1: In accordance with your suggestions, this topic has been dealt with in the discussion section (lines: First paragraph (lines:260-261-262) (However, when the agreement between the new classification criteria and the revised Sapporo classification criteria was assessed in our cohort, the performance of both criteria was statistically significant and significantly concordant.) (lines:345-346-347-348-349-350-351-352-353-354-355-356-357-358) (The sensitivity of the new criteria was lower in our cohort (73% and 62% (without livedo racemosa)) than in the 2023 ACR/EULAR APS classification criteria validation cohorts (83% and 84%, respectively), mainly due to seronegative APS patients (n = 4), non-persistent positivity of the LAC test (n = 5), and patients with obstetric involvement who did not meet the newly defined clinical criteria score. When we analysed the five patients with non-persistent LAC tests, we found that anticoagulant treatment was initiated after the clinical findings and LAC positivity, and the control LAC test was not accepted as persistently positive because it was positive under anticoagulants. In the scoring system of the new criteria, one point for ≥ 3 consecutive losses before 10 weeks' gestation and the fact that this score alone is not sufficient (at least three points are required) was one of the main factors that decreased the sensitivity in our cohort. In addition, the fulfilment of the new criteria in patients (n=7) in the no-APS group, in whom thrombocytopenia (two points) and livedo racemosa (two points) were detected, led to a decrease in specificity.)

(lines:359-360-361-362-363-364-365-366-367-368-369-370-371-372-373-374) (On the other hand, the performance of the newly published 2023 ACR/EULAR APS criteria and the revised Sapporo criteria were significantly compatible in our cohort. The main factors for these results seem to be the more difficult fulfilment of the new criteria due to the above mentioned assessment of obstetric involvement and the inclusion of findings such as thrombocytopenia and livedo racemosa, which were not included in the Sapporo criteria, in the new criteria. The fact that both criteria worked well and were compatible in our cohort could raise the question of what additional contribution the new criteria make compared to the revised Sapporo criteria. However, the inclusion of other clinical findings that may occur in the course of APS besides thrombosis and obstetric involvement in the new classification criteria may be important in order for rheumatologists to focus on other clinical findings. Although it is recognised that classification criteria should only be used for classification, they are also frequently used as diagnostic criteria. According to our study, the sensitivity of the new criteria in our cohort (73% and 62% (without livedo racemosa)) was low, and therefore, the limitations of the criteria should be considered when making a diagnosis. On the other hand, the specificity of the new criteria reached 100% when the livedo racemosa finding was not considered.)

and the abstract section (lines:13-14-15-16-17-18-19) (The performance (sensitivity, specificity and area under the curve (AUC)) of the 2023 ACR/EULAR classification criteria for APS was evaluated and the agreement with the revised Sapporo criteria was compared using the kappa test. In our cohort, the sensitivity and specificity of the 2023 ACR/EULAR classification criteria for APS were 73% and 94%, respectively (AUC: 0.836, 95% CI: 0.772-0.899), while the sensitivity and specificity of the revised Sapporo criteria were 66% and 98%, respectively (95% CI: 0.756-0.888). The performance of the two sets of criteria in our cohort was significantly consistent and significant (p<0.001).), (lines: 21-22-23) (Although the newly published criteria broaden the scope of APS classification by including clinical findings other than thrombosis and obstetric criteria, their sensitivity in our cohort was low.).

Comment 2: The differences in ROC, sensitivities and specificities should also be given in the abstract.

Response 2: The abstract section has been revised according to your suggestions (lines: 15-16-17-18-19). (In our cohort, the sensitivity and specificity of the 2023 ACR/EULAR classification criteria for APS were 73% and 94%, respectively (AUC: 0.836, 95% CI: 0.772-0.899), while the sensitivity and speci-ficity of the revised Sapporo criteria were 66% and 98%, respectively (95% CI: 0.756-0.888). The performance of the two sets of criteria in our cohort was significantly consistent and significant (p<0.001).)

Comment 3: It would be of interest to know what the diagnoses were in the 10 ‘additional IRD patients’.

Response 3: In the first paragraph of the ‘Results’ section of the article, the diagnoses of 10 patients are given (lines:124-125-126-127). (Among the patients in the no-APS group (others, n=10) with an IRD diagnosis other than SLE and a history of thrombosis, Behçet's disease was the most common (n=6). Mixed connective tissue disease (n=1), rheumatoid arthritis (n=2), and Sjogren's syndrome (n=1) were the other IRDs in patients with a history of thrombosis.)

Comment 4: English language editing is needed. I do not understand what ‘could not be controlled’ in line 298 means.  

Response 4: The current sentence has been edited according to your suggestions (lines:380-381-382). (Another important limitation is that the livedo findings could not be assessed by re-examining the livedo reticularis and livedo racemosa due to the retrospective design of the study.)

English language editing for the full article was done by MDPI and the certificate is attached.

Round 3

Reviewer 3 Report

Comments and Suggestions for Authors

I believe that the raised issues have been addressed adequately.
